# Prevalence of Autoimmune and Autoinflammatory Diseases in Chronic Urticaria: Pathogenetic, Diagnostic and Therapeutic Implications

**DOI:** 10.3390/biomedicines11020410

**Published:** 2023-01-30

**Authors:** Giuseppe Murdaca, Francesca Paladin, Matteo Borro, Luisa Ricciardi, Sebastiano Gangemi

**Affiliations:** 1Department of Internal Medicine, University of Genoa, 16132 Genoa, Italy; 2Ospedale Policlinico San Martino IRCCS, 16132 Genoa, Italy; 3Internal Medicine Department, San Paolo Hospital, 17100 Savona, Italy; 4Department of Clinical and Experimental Medicine, School and Operative Unit of Allergy and Clinical Immunology, University of Messina, 98125 Messina, Italy

**Keywords:** chronic spontaneous urticaria, autoimmune diseases

## Abstract

Chronic spontaneous urticaria (CSU) is defined as the almost daily occurrence of widespread wheals, angioedema, or both, for more than 6 weeks. It affects 1–2% of the general population, with a higher prevalence in female patients, and is more frequent patients over 20 years of age. More than half of all cases of chronic idiopathic urticaria are thought to occur due to an autoimmune mechanism, specifically the production of autoantibodies against the high-affinity immunoglobulin E (IgE) receptor (FcεRI). The quality of life in these patients is often greatly compromised, also due to the onset of comorbidities represented by other autoimmune diseases, such as thyroid disease, rheumatoid arthritis, systemic lupus erythematosus, Sjögren’s syndrome, celiac disease, and type 1 diabetes, among others. This review aimed to analyze the close correlation between CSU and some autoimmune and autoinflammatory diseases, in order to encourage a multidisciplinary and multimorbid approach to the patient affected by CSU, which allows not only control of the natural course of the disease, but also any associated comorbidities.

## 1. Chronic Urticaria

Chronic spontaneous urticaria (CSU) is defined as the almost daily occurrence of widespread wheals, angioedema, or both for more than 6 weeks [1]. It affects 1–2% of the general population, and it is more frequent in women and in patients older than 20 years [2,3,4]. Quality of life is often highly compromised, with interference in daily routine and consequent onset of psychiatric comorbidities, such as depression and anxiety [5,6]. Symptoms of chronic urticaria comprehend hives, angioedema or both (in almost one-third of patients). Clinical manifestation may range in frequency and duration: about 50% of patients have symptoms for 3–6 months with a self-limiting evolution, whereas more than 10% of patients present symptoms for 5 years or more [2,7,8]. Classification of urticaria is made according to the triggers and patient-reported signs and symptoms, and it may be divided into spontaneous and inducible [1]. This review aimed to analyze the close correlation between CSU and some autoimmune and autoinflammatory diseases, in order to encourage a multidisciplinary and multimorbid approach to the patient affected by CSU, which allows not only control of the natural course of the disease, but also any associated comorbidities.

### 1.1. Physiopathology

New findings in urticaria showed that it is not only a mast cell-driven disease, but also basophil degranulation and other immune cells like eosinophils, T and B lymphocytes, epithelial, and endothelial cells are involved [9]. Tissue mast cells can be activated by different triggers, both immunological and non-immunological. Among the immunological activation of mast cells, there is the well-known crosslinking of allergens recognized by IgE molecules attached to high-affinity receptors (FcεRI) on the membrane, but other mechanisms are also known to be involved, like C5aR for anaphylotoxins C5a, CRTh2 for PGD2, MRGPRX2 for neuropeptides (like substance P), or proteases and cationic proteins (like MBP and ECP), cKit for stem cell factor (SCF), cytokine receptors like IL-4Rα and TSLP-R, different kinds of TLRs to recognize PAMPs and DAMPs, and also some inhibitory receptors like Siglec-8 to downregulate the activation pathways [10]. Among non-immunological pathways of activation, there is the coagulation pathway that induces mast cell degranulation through protease-activated receptors (PAR) with thrombin and D-dimer formation. Moreover, recently a dysregulation of intracellular signaling pathways in mast cells and basophils leading to defect in cell function and the production of IgG autoantibodies directed against FcεRI or IgE on both mast cells and basophils or IgE autoantibodies against autoantigens like thyroid peroxidase, DNA and IL-24 have been demonstrated to be involved in urticaria pathogenesis [11]. Regardless of the mechanism, the release of histamine, platelet-activating factor, tryptase, leukotrienes, and other cytokines lead to sensory nerve activation, vasodilatation, and plasma extravasation, as well as cell recruitment typical of urticarial lesions [10]. Each different possible pathway of activation correlates to a different endotype of urticaria that produce different and similar phenotypes.

### 1.2. Two Endotypes of CSU

Recently, two different endotypes of CSU have been proposed: (1) IgE-mediated CUS, also known as autoallergic CSU and (2) type IIb autoimmune CSU [12,13,14,15], characterized by the presence of IgG autoantibodies, and probably IgM and IgA that are responsible for direct activation of mast cells with the binding of high-affinity IgE receptors [16,17].

The diagnosis of type IIb autoimmune CSU is based on a combination of positive autologous serum skin test, basophil histamine release assay and/or basophil activation test and immunoassay for specific IgG autoantibodies against FcεRIα/IgE [18]. Among patients diagnosed with CSU, it has been estimated that 8% have this endotype of urticaria that usually presents with low levels of IgE and a higher rate of eosinopenia and basopenia [12,19]. In 2021, Kolkhir et al. [20] showed that patients diagnosed with CSU classified into a type IIb autoimmune CSU had higher frequency of other autoimmune diseases when compared to type I CSU. Specifically, patients with a positive autologous serum skin test, basophil histamine release assay and basophil activation test had a risk 1.7, 2.9 and 3.3 times greater of having other autoimmune diseases, respectively.

## 2. Associations with Other Autoimmune Diseases

Chronic urticaria has been always described with and correlated to the presence of various other pathologies, including autoimmune, psychiatric, and atopic diseases [1]. In particular, the prevalence of autoimmune diseases in CSU is increased, e.g., ≥1% in most studies compared to ≤1% in the general population [21]. Autoimmune diseases can be divided into organ-specific and non-organ-specific diseases [22,23]

### 2.1. Organ-Specific Autoimmune Disease and CSU

In a metanalysis on 60 studies, the prevalence of organ-specific autoimmune diseases in patients diagnosed with chronic spontaneous urticaria reached 27.5%. The most common organ-specific autoimmune disease in CSU were endocrine, especially Hashimoto thyroiditis, followed by hematological, like pernicious anemia, and skin, like vitiligo [21]. These results were then confirmed by the authors in a further retrospective study in 2021, with 28% of patients with urticaria that presented at least one autoimmune disease [20].

#### Thyroid Autoimmunity

Among autoimmune disease, thyroid autoimmune disease has been linked to Chronic Urticaria for decades. In 2009, a study performed by Gangemi S. et al. found that one-third of patient diagnosed with chronic idiopathic urticaria were positive for at least one thyroid autoantibody. This high rate was one of the highest when compared to other study performed in that period [24]. A further review on the topic showed the presence of some reports assessing a possible connection, supported by improvement of symptoms when patients started thyroid substitutive therapy [25]. A recent metanalysis has been published by Kolkhir et al. [26]. Specifically, they found a strong correlation between CSU and high levels of autoantibodies anti-thyroid of G class, especially IgG against thyroid peroxidase. This correlation was present more frequently in adults than in children. Despite higher levels of IgE against thyroid peroxidase has been found in patients suffering from chronic urticaria than in control, this correlation presented weak evidence. Moreover, in line with previous findings, thyroid disfunction was strongly correlated with the onset of CSU, especially hypothyroidism and Hashimoto’s thyroiditis. These results are in agreement with a previous study on 12,778 patients with a diagnosis of chronic urticaria [27]. The correlation between presence of thyroid autoimmunity and urticaria showed a weak strength, thus leading to a reduced importance of thyroid autoantibodies evaluation at first assessment of urticaria in recent guidelines [28]. Finally, authors found that treatment of hypothyroidism, like the administration of levothyroxine, was strongly correlated with an improvement in urticaria symptoms [26]. A similar effect was also shown in a report of chronic urticaria and Graves’ disease, in which the improvement of hyperthyroidism was correlated with a marked improvement in urticaria symptoms [29]. Kolkhir et al. concluded their research speculating about possible pathogenic mechanisms that may sustain their findings, including that IgG thyroid autoantibodies may enhance mast cells’ susceptibility to other activating signals [30]; IgE thyroid autoantibodies may induce formation of immune complexes and subsequent activation of the complement that may trigger urticaria [26,31,32].

A recent systematic review and meta-analysis on 19 case-control studies further evaluated the overall risk of thyroid autoimmunity in patients diagnosed with CU [33]. Authors showed that the diagnosis of CU was associated with an approximately fivefold higher risk of presenting positivity for anti-thyroperoxidase antibodies, a marker of chronic autoimmune thyroiditis. They speculated two possible mechanisms that may coexist. One mechanism comprehends the involvement of IL-6, which increases endothelial permeability in an in vitro study [34] and has been linked to urticaria flares: it increases in patients with ongoing urticaria and reduces when urticaria symptoms go into remission [35]. Interestingly, high levels of circulating IL-6 have been found in Hashimoto’s thyroiditis and positively correlate with the number of Th22 lymphocytes that are related to anti-thyroperoxidase antibodies [36,37]. The second speculated mechanism regards a possible deficiency in the activity of regulatory T cells (T regs), which are characterized by suppressive functions on effector immune cells, and their decrease in number or function has been associated with autoimmune disorders [38]. Interestingly, among disorders linked with these alterations, there are both CU [39,40,41] and autoimmune thyroid diseases [42,43]. However, the meta-analysis presented some limitations, including the analysis of only anti-thyroperoxidase antibodies, which are not the unique auto-antibody of autoimmune thyroid diseases and, in contrast, are not sufficient to make diagnosis of thyroid autoimmunity (they can be found in subacute thyroid autoimmunuty and in non-autoimmune disorders) [44].

### 2.2. Type I Diabetes Mellitus

A study evaluating 12,778 patients with chronic urticaria showed that type I diabetes mellitus had an odds ratio of 7703 compared with that in control subjects. Dividing the study population into male and female, authors showed a higher odds ratio in female (12.92, 95% CI, 6.53–25.53; *p* < 0.0005) than in men (2.34; *p* = 0.015). The onset of type I diabetes mellitus was after years from CU diagnosis.

The systematic review performed by Kolkhir et al. [21] on five previous studies showed that prevalence of insulin-dependent diabetes mellitus is 0.5% in patients diagnosed with chronic spontaneous urticaria, ranging from 0.2 to 5.5%.

A recent study performed with the aim of evaluating the presence of urticaria among young patients diagnosed with type I diabetes mellitus showed interesting findings [45]. A total of 5895 participants were collected, with a ratio 1:4 between case and control group respectively. The incidence rate of urticaria in patients with T1D was 26.6 per 1000 person-years in contrast with 6.85 per 1000 person-years in patients without T1D. The hazard ration of urticaria was 2.84 (95% CI = 2.27–3.56). In females diagnosed with T1D, the risk of urticaria was 2.05 (95% CI = 1.50–2.80) higher than for those without T1D, and the relative risk in male patients was 4.35 times higher (95% CI = 3.08–6.16). The authors calculated that the risk of urticaria in younger people diagnosed with T1D was 3.62 higher than in those without T1D. Moreover, they found that, when the score of the adapted Diabetes Complications Severity Index was over 1.01, there was a significant increase in the risk of urticaria. Specifically, when the adapted Diabetes Complications Severity Index was between 1.01 and 2.00 per year, the hazard ration of urticaria was 2.57 (95% CI = 1.18–5.57), and when the score was over 2.00, the risk increased to 4.47 (95% CI = 2.68–7.47). The authors assumed that fluctuation of glucose levels, especially a reduction, could activate mast cell degranulation and thus trigger urticaria [45]. To further support these findings, some case reports describe the association between T1D and CU [46,47]. An interesting case report by Jesenak et al. [48] described a patient with diagnosis of type 1 diabetes mellitus and autoimmune thyroiditis who started to present urticaria, with negative work-up. After diagnosis of CSU and the failure of first-line treatment, the patient was treated with Omalizumab. Interestingly, Omalizumab not only obtained a complete recovery from CSU, but also achieved control of diabetes mellitus with reduction in both in glycemic control and glycated hemoglobin. This report can induce speculation about a possible IgE-mediated mechanism underlying these two conditions. A similar result was described by Del Barba et al. [49]: a 13-year-old patient with chronic spontaneous urticaria and type 1 diabetes mellitus was successfully treated with Omalizumab without worsening their glycemic control, in contrast with previous large-scale studies that suggested an increase in insulin resistance due to this therapy [50,51,52]. This difference can be due to a possible beneficial effect of Omalizumab on T1D that may cover the hyperglycemic effect seen in type 2 diabetes mellitus [50,51,52].

### 2.3. Celiac Disease

A study conducted on 12,778 patients with a diagnosis of CU and 10,714 control subjects, spread over 17 years, showed a certain association between CU and celiac disease [27]. Specifically, in patients diagnosed with CU when compared to controls, the odds of having celiac disease was 26.96 overall (CI 6.6–110.17). When analyzing by sex, women presented higher odds: 57.83 CI 7.99–418.29 for women and 3.90 0.50–30.27 for men. Moreover, the authors found that celiac disease was diagnosed mostly after CU: 17.2% and 82.2%, respectively.

A systematic review showed that celiac disease is the third autoimmune disease in order of overall prevalence associated with CU, after rheumatoid arthritis and Hashimoto’s thyroiditis [21]. The papers evaluated by the authors were dated from 1992 to 2015 and the prevalence of celiac disease in association with CU ranges from 0.5 to 9.3. The calculated mean prevalence for celiac disease among CU diagnosis has been estimated to be 0.7%.

Another study, performed by Ludvigsson et al. [53], confirmed a possible association between urticaria, chronic urticaria and celiac disease. The authors showed hazard ratios of 1.51 for any urticaria (95% CI = 1.36–1.68) and 1.92 for chronic urticaria (95% CI = 1.48–2.48) among 28,900 patients histologically diagnosed with celiac disease. The absolute risk for urticaria and chronic urticaria in the population evaluated was 140/100,000 and 24/100,000 person-years, respectively. Moreover, the authors found that patients with celiac disease had a certain risk of developing urticaria or chronic urticaria prior to the celiac disease diagnosis (odds ratio, OR = 1.31; 95% CI = 1.12–1.52 and OR = 1.54; 95% CI = 1.08–2.18, respectively).

The eventual association between these two conditions is further supported by some case reports. Haussmann et al. [54] described a 24-year-old woman with intermittent urticaria and gastrointestinal complaints that resolved after histological diagnosis of celiac disease and adherence to a gluten-free diet. Peroni et al. [55] described a subclinical case of celiac disease who presented with chronic urticaria. In agreement with previous reports, Sanchez et al. [56] showed a case of a 5-year-old boy who presented cold urticaria and, suddenly, was diagnosed with celiac disease without gastrointestinal manifestations. Cutaneous symptoms improved with gluten dietary avoidance. Candelli et al. [57] and Scala et al. [58] showed similar results in their respective case reports. In contrast, Levine et al. [59] described a patient with CU and chronic urticaria that did not improve with food avoidance. A similar result was described by Mennini et al. in another case report that presented anaphylaxis with urticaria as a symptom after ingestion of accidental wheat, being previously diagnosed with celiac disease [60]. Meneghetti et al. [61] described a case series of 32 urticaria patients, in which three were diagnosed with celiac disease, and thus suggested to screen for celiac disease in patients diagnosed with chronic urticaria.

A case-control study performed by Caminiti et al. [62] showed that diagnoses of chronic urticaria were statistically more frequent in children with celiac diseases (5.0%) when compared with controls (0.67%) (*p* = 0.0003). To further support the association, they showed that gluten-free diet resulted in urticaria remission.

An interesting case report described by Wong et al. [63] showed an association among celiac disease and subsequent onset of IgE-mediated hypersensitivity reaction to wheat. This may induce speculation about a possible association between these two conditions, despite a different pathogenesis.

### 2.4. Vitiligo

In the systematic review performed by Kolkhir et al., vitiligo has been classified as the sixth more frequent autoimmune disease associated with chronic urticaria. Among nine different studies analyzed, the prevalence of vitiligo in patient diagnosed with CSU has been estimated to be 0.6–9.8 [21]. The prevalence of urticarial rush in patients diagnosed with vitiligo has been estimated to be 0.5–1 among three different studies [21]. In 2021, a retrospective study on 1199 CSU patients found that vitiligo was the second most frequently associated condition with CUS among autoimmune diseases, after autoimmune thyroiditis. The estimated prevalence has been reported to be 2.3% [20]. Moreover, some case reports and a case series described the concomitant presence of vitiligo and urticaria in patients [64,65,66].

### 2.5. Pernicious Anemia

Magen et al. [67], Mete et al. [68], and Leznoff and Sussman [69] demonstrated a certain association between low levels of vitamin B12 caused by autoimmune production of antibodies and chronic urticaria, with a prevalence of 5.4, 6.1, and 5.5, respectively. The overall prevalence of Pernicious Anemia has been calculated to be 0.2% [21].

#### Addison’s Disease

The association between autoimmune adrenalitis and/or Addison’s disease and chronic urticaria is not so strong as in previous diseases. In two different studies, the prevalence of the disease in patients suffering from chronic urticaria ranges from 0.6 to 2.2 [26,69].

## 3. Non-Organ Specific Autoimmune Diseases and CU

Taking together all systemic autoimmune diseases, the most frequent correlation with urticaria has been seen in connective tissue diseases, with rheumatoid arthritis placed first [21]. The onset of a systemic autoimmune disease has been demonstrated with a higher incidence in female patients with urticaria, especially for rheumatoid arthritis (RA), Sjögren’s syndrome (SS), and systemic lupus erythematosus (SLE). Moreover, the presence of a positive rheumatoid factor and antinuclear antibodies were significantly more frequent in patients with chronic urticaria [27]. Ghazanfar et al. [70] have in fact highlighted how the increased general inflammatory state caused by CU correlates significantly with the increased prevalence of rheumatoid arthritis in women affected by CU (HR = 1.8), as a consequence of mast cell destabilization. In addition, UC has been shown to have a genetic association with the human leukocyte antigen (HLA)-DR4 and HLA-DQ8 alleles, of which the former is strongly associated with rheumatoid arthritis [71]. A cross-sectional study of 390 CSU patients reported a higher prevalence of connective tissue diseases (CTDs) such as RA, SS, and SLE in UC patients than in the normal population, with a prevalence of 1.8% for RA and of 0.3% for SLE [72]. Chiu et al. have deepened the biochemical correlation existing between CU and SLE, recognizing a fundamental role of autoantibodies, complement activation, and the coagulation cascade. CU and SLE were in fact both characterized by a higher prevalence of autoantibodies against FceRI and IgE. The appearance of urticaria in patients with active SLE is due to immune complex deposition and complement activation; this could indicate the key role of SLE in the etiopathogenesis of UC. Furthermore, an increased risk of SLE with UC was confirmed only in women, particularly in those aged between 20 and 59 years [73,74,75]. CSU can also be triggered by drugs routinely used to treat lupus, especially nonsteroidal anti-inflammatory drugs, with a predominance of constitutional, mucocutaneous, and musculoskeletal involvement, without a high frequency of manifestations of severe lupus [76].

## 4. CU and Other Immune-Mediated Diseases

A recent study by Magen et al. highlighted a high prevalence of UC in patients affected by alopecia areata (AA), recognizing in the IgG-anti-IgE autoantibodies and RI anti-Fc (high affinity IgE receptor) the possible physiopathological correlation factors [77]. The presence of such functional FceRIa-Ab anti-IgE antibodies in UC patients is responsible for the release of histamine by activating the complement system. The same immunological mechanism is also found in subjects affected by myasthenia gravis (MG), an autoimmune disease that causes weakness of striated muscles mediated by the AChR-Abs antibody, responsible for the degradation of the neuromuscular junction by activation of the complement system. Although the pathogenic link between these two autoimmune diseases remains unclear, complement activation plays an important pathophysiological role in both diseases, being able to represent their possible link [78,79].

The international EAACI/GA^2^LEN/EuroGuiDerm/APAAACI guideline for the definition, classification, diagnosis, and management of urticaria [28], published in March 2022, reports that it is well known that causes of CSU include autoimmunity type I (CSU)aiTI, or “autoallergic CSU”; with IgE autoantibodies against autoantigens and type IIb autoimmunity (CSU)aiTIIb; with activating autoantibodies directed by mast cells. In the diagnostic process, the patient’s history and a physical exam can provide clues to the underlying causes. For this purpose, the execution of basic tests, such as CRP (more often elevated) and the dosage of eosinophils and basophils (more often reduced in CSUaiTIIb), is of primary importance. Testing for TPO-IgG and total IgE should also be performed in CSU patients in specialist care, as they are useful for diagnostic purposes. CSUaiTIIb patients are more likely to have low or very low total IgE and elevated anti-TPO IgG levels, and a high ratio of anti-TPO IgG to total IgE is currently the best surrogate marker for CSUaiTIIb. Other underlying causes include active thyroid disease, infections, inflammatory processes, food and medications. In CSU, the most common comorbidities are CIndUs, autoimmune diseases, and allergies. Findings from the patient’s medical history, a physical exam, or baseline tests that indicate a comorbidity or consequence of CSU should prompt further investigation, such as screening for specific diseases by means of questionnaires, provocation tests, further laboratory tests, or referral to a specialist.

## 5. CSU and Autoinflammatory Diseases

Autoinflammatory diseases (Cryopyrin-Associated Periodic Syndrome (CAPS), Familial Mediterranean Fever (FMF), Schnitzler Syndrome (SchS), Adult Still’s Disease (AOSD)) represent a group of chronic disabling diseases characterized by a condition of direct self-inflammation, mediated by disturbances in innate immune signalling pathways [80]. Specifically, an excessive secretion of cytokines by innate immune cells (such as macrophages, monocytes) is observed, among which we mention the group of interleukin (IL)-1, which, accumulating at the level of different tissues and systems, lead to the clinical development of disease. The latter includes recurrent fever attacks, musculoskeletal, gastrointestinal, and neurological involvement. The integumentary system is also a frequent site of autoinflammatory disease, with typical symptoms including urticaria, pustular and ulcerative lesions [81].

The urticarial rash in patients with autoinflammatory syndromes is often grossly indistinguishable from that in patients with CSU. However, it is possible to observe, in these cases, a wider spectrum of lesions than true urticaria, i.e., flat wheals that can, at first sight, resemble erythematous patches, but also more solid and stable lesions. The distribution of this type of urticarial rash is also slightly different, being distributed rather symmetrically over the trunk and/or extremities, usually sparing the head [82].

Considering the great clinical similarity between these two diseases, the need arises, where an autoinflammatory disease is suspected, to perform specific tests, including dosage for elevated inflammatory markers; serum protein electrophoresis to rule out monoclonal gammopathy in adults; urinalysis to screen for proteinuria from secondary renal amyloidosis; and skin biopsy to look for neutrophil-rich infiltrates. If an inherited autoinflammatory disease is suspected, testing for mutations in the relevant genes should be performed [83].

Within the broad spectrum of autoinflammatory diseases, two are those that present most frequently clinically with urticarial lesions that enter into the differential diagnosis with CSU: urticarial vasculitis (UV) and Schnitzler Syndrome (SchS).

If a form of UV is suspected, the differential diagnosis with CSU [84] is based on testing for skin histopathology, as well as specific laboratory tests, such as complete blood count, serum creatinine, C-reactive protein (C-RP), rate of erythrocyte sedimentation (ESR), urinalysis, complement studies (C1q, C3, C4), and anti-C1q antibody analysis underlying connective tissue disease or viral infection [84,85].

As for SchS, a member of the family of autoinflammatory syndromes, this too often manifests itself with chronic recurrent, nonpruritic and confluent plaques. These urticarial plaques can persist for up to 24 h and then resolve without residual pigmentation. Characteristic of SchS is the presence of monoclonal paraproteinemia, typically IgM, with neutrophil-dominated cellular infiltrates and upregulation of cytokines and inflammasome components in lesional skin, supporting them as potential biomarkers to differentiate SchS from CSU [86]. IL-1-related cytokines and inflammasome components are upregulated in skin biopsy specimens from SchS patients, resulting in products from mast cells (IL-1β, IL-6) and neutrophils (IL-6, IL-18). In view of these immunohistochemical features, it is recommended to apply a panel of skin biomarkers, including MPO, IL-1β, IL-6, IL-18, as well as ASC and caspase-1, to differentiate SchS from CSU [85].

## 6. Conclusions

CU is supported by recognized immunopathological mechanisms, in which autoantibodies against the high affinity IgE receptor play a fundamental role in the activation of mast cells, basophils and the complement system, responsible for the appearance of clinical signs of the disease and its reactivation. Due to its immune-mediated nature, CU has a wide range of associated comorbidities, in which autoimmune diseases are undoubtedly the most frequent (Figure 1) [20,21,27,72]. In particular, autoimmune thyroiditis, type I diabetes and celiac disease have shown the strongest correlation. The latter are associated, albeit to a lesser extent, with some non-organ-specific autoimmune diseases, such as rheumatoid arthritis, Sjögren’s syndrome, and systemic lupus erythematosus. In a smaller percentage of cases, the presence of cutaneous urticarial lesions can represent the clinical manifestation of autoinflammatory diseases; for this reason, the patient’s anamnestic history and specific batteries of tests are fundamental in the differential diagnosis of the different forms of CSU (Figure 2).

Although the data currently available in the literature are not sufficient to clearly define the pathophysiological correlation between CU and these autoimmune comorbidities, we believe it is important for the clinician who approaches the patient with CU to consider and investigate the possible presence of other immune pathologies. This approach makes it possible to direct the treatment towards a multidisciplinary and multimorbid approach, in order to control the natural course of the CU and any associated comorbidities.

## Figures and Tables

**Figure 1 biomedicines-11-00410-f001:**
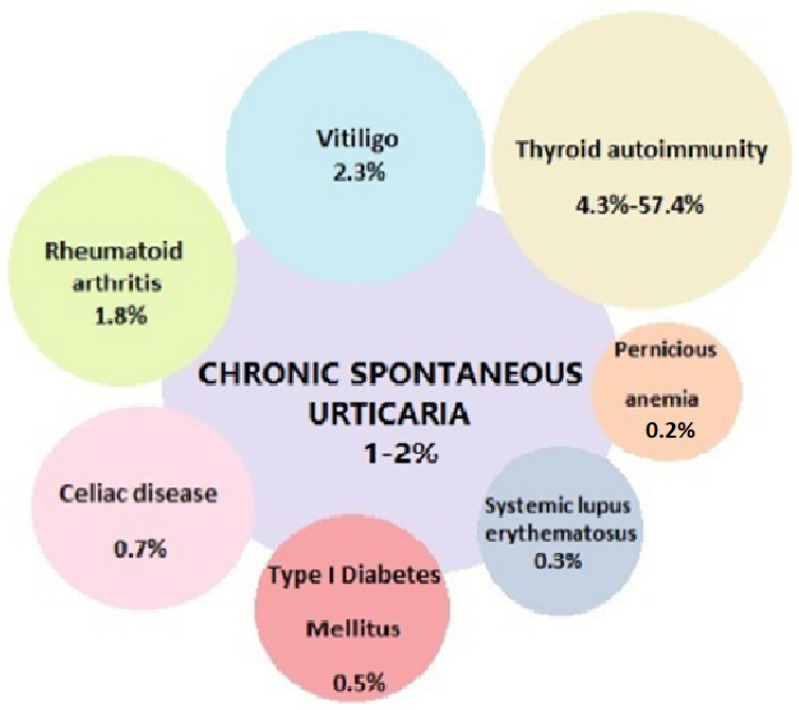
Prevalence of autoimmune diseases in CU.

**Figure 2 biomedicines-11-00410-f002:**
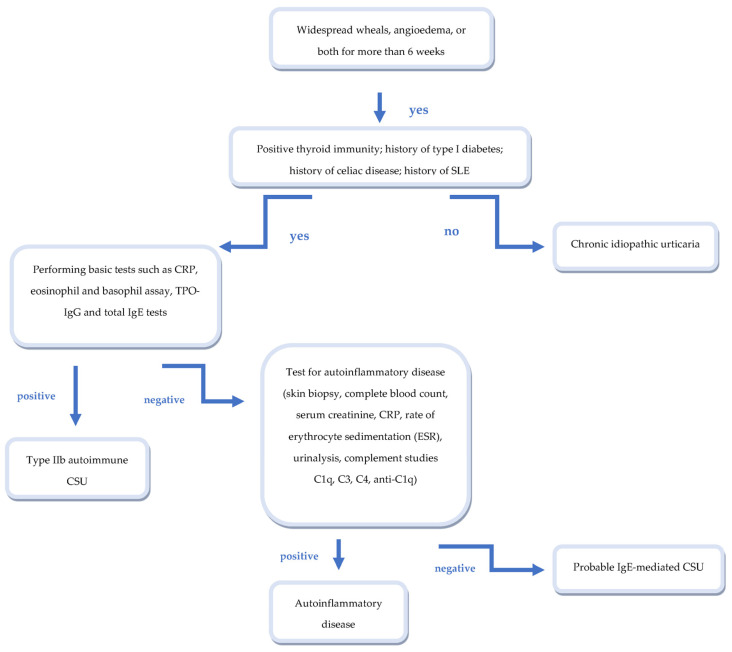
Algorithm for the differential diagnosis of urticarial rash.

## Data Availability

Not applicable.

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
