# Peer review of "Prevalence of Autoimmune and Autoinflammatory Diseases in Chronic Urticaria: Pathogenetic, Diagnostic and Therapeutic Implications"

_biomedicines, 2023, doi:10.3390/biomedicines11020410_

Round 1

Reviewer 1 Report

This is very important review on autoimmune diseases in chronic urticaria (CU).

For years chronic urticaria was regarded as allergic disease, which is not really the case and treatment modalities using biologic drugs demonstrate its efficacy. Presented manuscript demonstrates importance of autoimmunity in CU in a very concie and clear cut manner. Congratulations to the Authors', very important work, clinically.

Author Response

Dear Reviewer, many thanks for your suggestions. I have revised the paper accordigly to your indications.

Giuseppe Murdaca

Reviewer 2 Report

I had to revise the article entitled ” Prevalence of autoimmune diseases in chronic urticaria: patho-genetic, diagnostic and therapeutic implications” submitted for publishing in Biomedicines journal.

Chronic spontaneous urticarial is a chronic disease that affects 1-2% of the general population, interfering with the daily routine activity and favoring the onset of psychiatric comorbidities such as depression and anxiety. There are several possible pathways to activate the immune cells degranulation, with consequent release of mediators responsible for producing typical urticarial lesions. Each of these pathways activation correlates to a different endotype of urticaria that produce different and similar phenotypes. The present review aimed to analyze the close correlation between CSU and some autoimmune diseases, in order to encourage a multidisciplinary approach to the patient affected by CSU

The article needs some revision before publish it.

1.     In the abstract, the authors should rephrase the following sentence, the information is duplicated (higher prevalence means it is more frequent).

“It affects 1-2% of the general population, with a higher prevalence in female patients and is more frequent in women and…”

2.   The authors should announce the aim of the review in the introduction section. The aim is mentioned only in the abstract.

3.     Revise please the typing errors. (paragraph 1.3 higher instead of hogher, paragrapgh 2.2 improvement of hyperthyroidism instead of improvement hyperthyroidism, paragraph 3 taking instead of takin)

4.   Thyroid autoimmunity is a part of organ specific diseases, so I suggest to re-numbered it as 2.2.1.

5.   The figure is not mentioned in the text. The authors should mention the references for the aformentioned prevalences, or if they calculated the prevalence they should write it how.

6.   The authors should propose an algorithm of multidisciplinary diagnosis in CU based on the findings presented in the article.

Author Response

Dear Reviewer, many thanks for your suggestions. I revise the paper accordingly to your indications.

Giuseppe Murdaca
